# Development of a Clinical Prediction Rule for Treatment Success with Transcranial Direct Current Stimulation for Knee Osteoarthritis Pain: A Secondary Analysis of a Double-Blind Randomized Controlled Trial

**DOI:** 10.3390/biomedicines11010004

**Published:** 2022-12-20

**Authors:** Paulo E. P. Teixeira, Daniela R. B. Tavares, Kevin Pacheco-Barrios, Luis Castelo Branco, Eric Slawka, Julie Keysor, Virginia F. M. Trevisani, Doug K Gross, Felipe Fregni

**Affiliations:** 1The MGH Institute of Health Professions, Boston, MA 02129, USA; 2Neuromodulation Center and Center for Clinical Research Learning, Spaulding Rehabilitation Hospital and Massachusetts General Hospital, Charlestown, MA 02129, USA; 3Harvard Medical School, Boston, MA 02115, USA; 4Department of Evidence-Based Medicine, Brazilian Cochrane Centre, Federal University of São Paulo, Sao Paulo 04038-001, Brazil; 5Department of Geriatrics and Gerontology, Federal University of São Paulo, Sao Paulo 04021-001, Brazil; 6Unidad de Investigación para la Generación y Síntesis de Evidencias en Salud, Vicerrectorado de Investigación, Universidad San Ignacio de Loyola, Lima 15024, Peru; 7Department of Rheumatology, Santo Amaro University, Sao Paulo 04743-030, Brazil; 8Harvard T. H. Chan School of Public Health, Harvard University, Boston, MA 02115, USA

**Keywords:** clinical prediction rule, transcranial direct current stimulation, knee osteoarthritis, chronic pain, clinical trial

## Abstract

The study’s objective was to develop a clinical prediction rule that predicts a clinically significant analgesic effect on chronic knee osteoarthritis pain after transcranial direct current stimulation treatment. This is a secondary analysis from a double-blind randomized controlled trial. Data from 51 individuals with chronic knee osteoarthritis pain and an impaired descending pain inhibitory system were used. The intervention comprised a 15-session protocol of anodal primary motor cortex transcranial direct current stimulation. Treatment success was defined by the Western Ontario and McMaster Universities’ Osteoarthritis Index pain subscale. Accuracy statistics were calculated for each potential predictor and for the final model. The final logistic regression model was statistically significant (*p* < 0.01) and comprised five physical and psychosocial predictor variables that together yielded a positive likelihood ratio of 14.40 (95% CI: 3.66–56.69) and an 85% (95%CI: 60–96%) post-test probability of success. This is the first clinical prediction rule proposed for transcranial direct current stimulation in patients with chronic pain. The model underscores the importance of both physical and psychosocial factors as predictors of the analgesic response to transcranial direct current stimulation treatment. Validation of the proposed clinical prediction rule should be performed in other datasets.

## 1. Introduction

Knee osteoarthritis (OA) is the most common form of the disease, with around 14 million Americans having symptomatic knee OA [1]. The recent global prevalence and incidence estimations are approximately 16% and 203 per 10,000 person-years, respectively [2]. Chronic pain from knee OA can substantially impact the quality of life, affecting both physical and psychological health. Among the approximately 10% of people over the age of 55 with painful knee OA that has become disabling, nearly one quarter are severely disabled [3].

Pain is the most disabling symptom of knee OA and is an important predictor of disability [4,5]. While in the past, OA pain was often described as a secondary symptom of joint degeneration, current evidence underscores the complex central and peripheral mechanisms that underlie chronic pain in OA [6,7,8,9,10]. Signs of altered central nervous system processing, such as central sensitization [11], and impaired pain modulation, including impaired functioning of the descending pain inhibitory system (DPIS), are common among chronic knee OA pain sufferers [8,9,12,13,14,15]. Because of the complexity of the mechanism underlying chronic pain in knee OA, successful treatment remains an unsolved challenge for healthcare providers.

Current methods used to treat knee OA pain include both pharmacological and non-pharmacological strategies. Non-steroidal anti-inflammatory drugs are frequently used in this population and can lead to severe adverse effects [16]. In addition, opioids are increasingly used, also with severe several side effects and a risk of opioid dependence [17,18,19,20]. Available non-pharmacological treatments such as exercise therapy, patient education, or cognitive-behavioral therapies can be effective but have, at best, modest effects on pain [21,22,23,24]. Ultimately, total joint arthroplasty may be necessary, elevating the already excessive economic burden of OA. Despite good functional results, approximately 20% of individuals still continue to report pain six months or more after surgery [25], probably due to remaining disturbances in the pain processing mechanisms.

Transcranial direct current stimulation (tDCS) is a non-invasive brain stimulation (NIBS) technique that has been shown to provide pain relief in chronic pain conditions, including knee OA [26], fibromyalgia [27,28,29,30], and chronic pain after spinal cord injury [31,32]. It is designed to act on the neuronal state to facilitate the modulation of dysfunctional excitability patterns in the brain and induce neuronal plasticity. Clinical trials have explored the analgesic effect of tDCS on knee OA [33,34,35,36,37,38] and have also demonstrated the efficacy of tDCS in reducing pain in other chronic pain conditions after the stimulation of brain regions related to pain processing [39,40,41,42,43,44,45,46,47,48,49,50,51,52]. However, although there have been positive and significant results in tDCS trials for pain management, the results are heterogeneous.

The existence of subgroups inside the knee OA population has been raised as a potential source of treatment response variability [53,54,55,56,57,58]. One potential strategy to diminish this variability is the development of clinical prediction rules (CPR). CPRs are widely used in rehabilitative medicine to assist in identifying subgroups of patients whose baseline characteristics indicate a high likelihood of responding to a particular intervention. Many CPRs exist in the field of musculoskeletal rehabilitation [59,60,61,62,63,64,65,66] but few exist for knee OA patients. This methodological strategy can be useful, as it can be directly applicable to individual patients. Variability in the treatment response is observed in chronic musculoskeletal trials targeting pain with NIBS [67,68]. Therefore, a CPR could assist both trialists and clinicians in better selecting patients to diminish this variability and ultimately achieve optimal outcomes with tDCS therapy.

CPRs are ultimately designed to improve clinical decision making. They consist of a combination of clinical and/or historical variables, collected during a baseline assessment, that best characterizes patients who have a high probability of success when administered a specific treatment protocol. A CPR indicates the accuracy, expressed as a sensitivity, specificity, or likelihood ratio (LR), of each individual predictor and of a group of the most potent predictors to identify patients whose response to treatment exceeds a certain threshold for success. CPRs have been used to create treatment-based classifications for patients that can be easily applied to clinical settings [59,61,62,63,64,65,66,69], and the research methodological standards used to develop CPRs are well established [70]. The first phase, derivation, involves the identification of factors with predictive power and require longitudinal data [53,61]. To the best of our knowledge, there are no treatment-based CPRs in the field of NIBS. Specifically, indications of tDCS for knee OA pain could benefit greatly from treatment-based CPRs as variability in the treatment response is not uncommon with this type of intervention [71].

The aim of this study was to develop a CPR that predicts a clinically significant short-term analgesic effect on knee OA pain after a tDCS treatment protocol. Defining a CPR for tDCS treatment success may improve the selection of patients for tDCS interventions and optimize tDCS treatment’s effectiveness for knee OA pain.

## 2. Methods

The current study is a secondary analysis of data from a published double-blind randomized controlled trial (RCT) that evaluated whether tDCS reduces knee OA pain in elderly individuals with a dysfunctional DPIS (NCT03117231). The parent RCT study was approved by the Human Research Ethics Committee of Sao Paulo Hospital (1685/2016) and was conducted according to the Declaration of Helsinki. All participants provided written informed consent. Procedures, including the participants, the intervention, and the primary and secondary outcome measures, are described elsewhere [13,36].

A summary of the RCT’s study procedures is provided in the Appendix A. The time point used to determine treatment success was at the end of the 3-week tDCS intervention protocol, as the short-term analgesic effect was the focus of this analysis.

### 2.1. Definition of Treatment Success

The Western Ontario and McMaster Universities’ Osteoarthritis Index (WOMAC) pain subscale was used to determine treatment success, and data were collected before (baseline) and after the 15 treatment sessions. Changes in the patients’ WOMAC pain subscale scores were categorized as success (responders) or non-success (non-responders) depending on whether a 25% pain reduction from baseline was achieved (representing a 5-point reduction in the raw 20-point WOMAC pain subscale score). This cutoff score was chosen on the basis of the minimal detectable change (MDC), the minimal clinically important difference (MCID), the standard error of measurement (SEM), and the minimum important change (MIC) values previously reported for knee OA (MIC = 4.18, MDC= 4.58, MCID = 2.2, SEM = 1.65—all based on the non-transformed 0–20 scale) [72,73,74].

### 2.2. Baseline Predictor Variables

To our knowledge, there are no CPR studies for the use of tDCS on chronic pain, and due to the exploratory nature of this derivation phase of the proposed CPR, all clinical variables collected at baseline for the main randomized trial were explored as potential predictive variables, irrespective of whether they had previously reported predictive relationships with tDCS treatment success or were supported by a clear theoretical framework. The details of the baseline predictor variables for the entire sample and for the “success” or “non-success” groups are displayed in Table 1, and details about the methodology used to collect them is provided in the Appendix A [13,75,76,77,78,79,80,81,82,83,84,85,86,87,88,89,90,91,92,93].

### 2.3. Analysis

Descriptive statistics were used to present the data’s characteristics in both the responder and non-responder groups. Central tendencies and respective variabilities were reported according to the data’s distribution.

All individual baseline variables were first dichotomized. This commonly used approach was used, as it is known to facilitate clinical understanding when interpreting the final prediction rule [94]. Due to the lack of well-established clinical cutoffs for all the explored variables, and to standardize the calculation procedures, cutoffs were defined using receiver operating characteristic curves. This process calculates the sensitivity and specificity for different cutoff scores that help in the construction of a visual graph that represents how well the predictor performs. Cutoffs were determined at the values where the highest possible area under the curve (AUC) could be achieved, which thus had the highest predictive power for treatment success [59,70,75]. The first step of the analysis involved creating univariate logistic regression models to determine the best potential predictive variables to be included in a multivariate logistic regression model to derive the final CPR [70,75]. All the dichotomized baseline variables were considered as independent variables, while the dichotomized WOMAC pain subscale variable was used as the dependent variable. Individual variables that had a regression coefficient significance of at least 0.2 (*p* ≤ 0.2) were retained for further analysis in the multivariate model. The AUC values from the dichotomization processes were analyzed to confirm if any of the pre-selected variables for the multivariate model had an AUC greater than at least 0.5. The accuracy statistics, including the sensitivity, specificity, and positive likelihood ratios and the respective 95% CI [95], were reported for each individual predictor variable. If any variable presented a low predictive power (AUC ≤ 0.5), they were excluded from the list of potential predictive variables to be explored in the multivariate model. All the variables considered for the multivariate model were then checked for dependence and collinearity by testing their interaction terms and correlations, respectively. If any variables were found to be colinear or dependent, the variable with the easiest and most practical clinical use was chosen to be considered in the multivariate model. Next, a multivariate logistic regression model was built using a manual backward elimination approach with a regression coefficient significance criterion of 0.05. Pearson’s chi-squared and Akaike’s information criterion were also used to determine the model with the best fit. The AUC for the final model was determined. The CPR was developed by examining the accuracy of the various groupings of the variables present in the final logistic regression model. The ultimate goal of the CPR was to include clinically sensible predictors that were practical and easy to collect, that would maximize the positive likelihood ratio, and that could correctly identify a worthwhile proportion of subjects who were positive and had treatment success [60].

## 3. Results

Of the 51 subjects who received the tDCS stimulation, and were therefore included in the analysis, 15 met our criteria for success (29%), while 36 (71%) did not. Their descriptive data are displayed at Table 1.

After dichotomization, 17 variables were associated with treatment success in the univariate logistic models and considered as potential predictors. Lequesne’s maximum distance walked score, Von Frey’s sensation assessment (hand), and the change in PPT after CPM (hand) were not considered for the multivariate modeling analysis, as they showed low discriminatory power (AUC less than 0.5). The accuracy statistics and their respective 95% CIs for these pre-selected dichotomized individual predictor variables can be seen in Table A1. After application of the backward elimination approach to construct the final multivariate model, the best final model included five clinical variables (Table 2).

The final model was statistically significant (chi-squared (5) = 34.67, *p* < 0.01), with an adequate goodness of fit (Pearson’s chi-squared (16) = 6.25; *p* = 0.985). All the regression coefficients were statistically significant (*p* < 0.05). In summary, the direction of the coefficients would mean that if subjects reported having a sleepiness level below 2.9 (0–10 VAS) on the VAMS sleepiness scale, a score lower than 52.59 on the MCS (SF12), a score lower than 37.87 on the PCS (SF12), a score lower than 10.5 on the Beck depression inventory, and a score lower than 43.5 on the WOMAC total score, they are more likely to improve with the tDCS intervention. The AUC used for model discrimination is depicted in Figure 1.

The best rule (Table 3) for predicting success had the presence of five clinical variables (positive LR, 14.40; 95% CI: 3.66, 56.69). Accuracy statistics were calculated for each of the predictor variables present. A subject who exhibited five of the criteria variables would have an 85% (95% CI: 60–96%) post-test probability of success, as opposed to four variables (79%; 95% CI: 54–92%). Twelve of the 14 subjects who were positive for five criteria were in the successful outcome group (Table 4).

## 4. Discussion

The current study presents the initial derivation step for the development of a CPR that predicts a positive and clinically significant analgesic effect on knee OA pain after a tDCS treatment protocol. To the authors’ knowledge, this is the first CPR ever proposed for tDCS treatment and for any neuromodulation intervention used to treat knee OA pain. The analysis presented here supports the idea that the rule is highly accurate and has relevant discriminatory properties to identify knee OA patients who will potentially benefit from tDCS treatment. This is a novel approach used in the field of neuromodulation.

We found that lower scores in the measures of psychosocial function (SF12 MCS and the Beck depression scale) were the most strongly predictive (largest odds ratio) of an analgesic tDCS response in knee OA patients with chronic pain and a dysfunctional DPIS. Previous studies have found that chronic pain conditions, including knee OA with dysfunctional DPIS, are associated with multiple alterations in psychosocial components such as depression, stress, and anxiety [95,96]. Similarly, a recent systematic review highlighted the influence of psychosocial components on the treatment response across multiple treatment modalities (e.g., pharmacological, physical therapy, and combined treatments [97]), showing that a higher burden of depression, anxiety, and pain catastrophizing reduced the likelihood of treatment success [98]. Our results are consistent with these findings.

Regarding the predictors of the response to tDCS, although several randomized controlled trials have shown positive effects in several conditions [99], only few studies have explored the predictors of response. For instance, Gunduz and Pacheco-Barrios et al. [100] found that phantom limb patients who responded to M1 anodal tDCS had more non-painful phantom symptoms (specifically movement sensations in the phantom limb) at baseline, but no psychosocial factors were detected as important predictors. On the other hand, Kambeitz et al. [101] found that negative affect and the number of depressive episodes were important predictors of treatment response for pre-frontal tDCS in patients with major depression. The heterogeneity in the predictors of tDCS may be explained by the different treatment protocols, such as different neural target brain areas (motor cortex versus pre-frontal areas) and population differences (musculoskeletal versus neuropathic pain populations). It can be argued that poor mental health status at baseline is an unspecific associated factor in an unfavorable response to any treatment, because the interference of affect induces less engagement with the therapeutic protocol (less target engagement and adherence to the protocols) [102]. However, interestingly, in our study, when we applied the prediction model to the participants who received sham tDCS (*n* = 53), our set of predictors did not discriminate between responders versus non-responders, and the model was not significant. Thus, our prediction rule does not seem to be associated with a placebo response. These results underscore the importance of the emotion-related domains as potential predictors of response to analgesia, especially for treatments targeting sensorimotor plasticity (such as M1 anodal tDCS). Nevertheless, the role of psychosocial factors as effect modifiers or mediators of some intervention modalities needs further exploration to understand the causal pathways of chronic pain treatments and to optimize the intervention design.

Our final predictive model included physical activity-related variables, such as the PCS (SF12) and the WOMAC total score. Our findings are similar to those of previous research in knee OA that has also found the same variables to be relevant for the process of decision-making process regarding treatment [103]. The presence of physical variables in our final model suggests that not only are the emotional components relevant in predicting a treatment’s success, but so are the physical aspects of health. Previous research has explored the relationship between different levels of physical activity with pain perception and central pain processing [104,105]. It has been suggested that higher levels of self-reported physical activity or health are associated with better function of the endogenous pain inhibitory system and lower self-reported pain, while the opposite is associated with a more sedentary lifestyle [105,106,107,108,109,110,111,112,113,114,115,116]. In fact, pain tolerance in chronic pain has been suggested to be more associated with physical activity than with emotional factors, such as depression and anxiety [104]. It is known that participation in regular physical activity offers global benefits in terms of motor cortex function, including an enhanced capacity for neuroplasticity and motor learning [109], and the evidence supports the concept of the motor cortex as an important modulator for chronic pain [110,111,112]. This suggests that physical function/activity may have relevant implications for pain regulation and the sensory aspects of pain. Additional research has also shown that motor-based interventions, such as mobilization with movement techniques, can affect the pain threshold and change cortical excitability [112,113,114]. In addition, although our study used only tDCS as the intervention, because tDCS can modulate neuronal excitability and impact the brain’s plasticity, the authors believe that motor-based interventions, (e.g., exercise) when combined with tDCS may have a synergistic effect on the pain processing areas of the brain. A previous systematic review has even recommended the combination of these two treatment strategies for improving pain in different MSK conditions [67].

Our results revealed a robust CPR with a percentage increase between the pre-test probability and the post-test probability of 56% (29% to 85%) for five variables, and 54% when two variables were present. The authors have not found any CPRs of neuromodulation treatments for knee OA for comparison. However, a few other rehabilitation-related research have proposed the use of CPR for other interventions used for knee OA pain. Currier and colleagues [115] developed a five-variable CPR to identify patients with knee OA pain with favorable short-term response to hip mobilization and reported a post-test probability of 97% (PLR = 12.9) when two variables were present. This study also reported a high pre-test probability of 68%, which makes the percentage of increase in probability (pre- to post-test) 24%, consequently limiting the value of the CPR for clinical decision-making. The difference in the pre-test probability between our study and that reported by Currier et al. likely exists because of the differences in how treatment success was defined and the unique characteristics of our sample (impaired descending pain inhibitory system). Amano et al. [116] reported a two-variable CPR with a PLR of 17.8 and a percentage of increase in probability of 59%, However, the derived CPR was used to determine the risk of falling in community-dwelling individuals with knee OA and it was not a treatment CPR. Our choice of using the five-variable CPR is indeed not definite. Excellent levels of post-test probability and PLR could also be achieved with fewer variables. The authors have suggested the five-variable CPR, considering the feasibility of performing the clinical tests and self-reported measures. As this CPR becomes further validated, clinicians are encouraged to consider the applicability of the CPR in their own clinical settings and how frustrating and/or debilitating it would be for their patient if the treatment was not successful.

This study has limitations. Since this study refers to the derivation phase of a CPR, the objective was to identify suitable variables to be included in a CPR and not to establish definitive criteria to predict treatment success. Our limitations include the relatively small sample size. Our model using five predictors is also likely to be overfitted to our sample. We were limited to the variables collected in the parent RCT; thus, other factors, such as chronicity or compartmental distribution of OA, could have been significant for predicting treatment success. The variables and their cutoff points need to be further validated in other studies, preferably with larger sample sizes. Furthermore, our sample of knee OA patients was composed of subjects previously identified as having a dysfunctional pain inhibitory system; therefore, our findings are potentially not applicable to a broader population. Another limitation is the fact that success was determined on the day of the last session after three weeks of treatment, so our proposed CPR was not appropriate for predict long-term improvements. In addition, we collected data in the context of the controlled environment of a clinical trial, and there may be other variables that are better suited for predicting success in clinical settings. Considering our limitations, any definitive use of the proposed CPR is considered premature.

## 5. Conclusions

This is the first CPR ever proposed for the neuromodulation treatment of tDCS. Our results suggest that both physical and psychosocial health-related variables are likely relevant to predict treatment success. Our findings present a robust first methodological step that could positively impact knee OA care as further validation of the proposed rule can help identify patients that will best respond to the tDCS intervention. Future research efforts are encouraged to refine and validate the proposed CPR, ideally using larger samples and prospective cohort study designs.

## Figures and Tables

**Figure 1 biomedicines-11-00004-f001:**
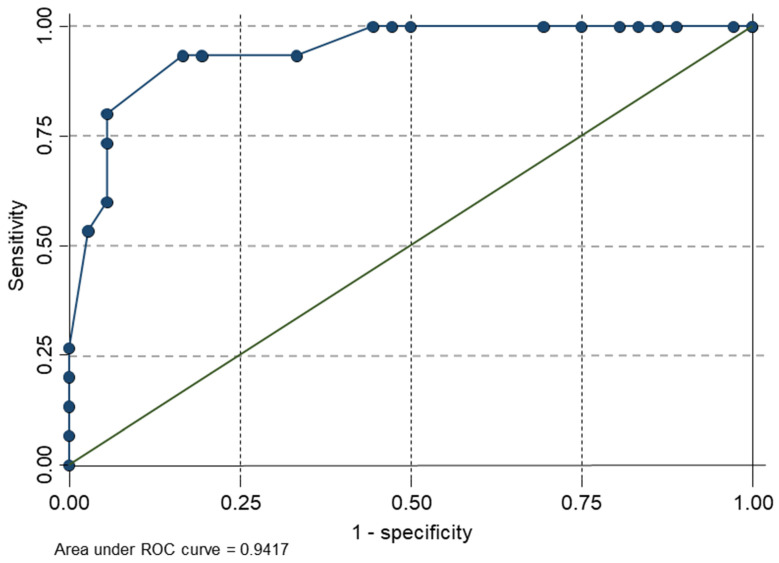
Graph of sensitivity versus 1 minus the specificity of the final predictive model.

**Table 1 biomedicines-11-00004-t001:** Variables assessed at baseline.

	All Subjects (n = 51)					Success (n = 15)			Non-Success (n = 36)			
Variables												
**Demographics**												
Gender—female; n (%)		42		82.35%		13		86.67%		29		80.56%
Age		74.8	±	7.44		75.33	±	8.17		74.6	±	7.23
**Pain-related variables**												
BPI pain severity		4.87	±	1.52		4.96	±	1.09		4.83	±	1.68
BPI interference—pain’s impact on function	*	4.57	IQR	3.1		3.78	±	1.89		4.8	±	2.02
VAS pain (now)	*	2.25		2.85	*	2.2		1.4	*	2.85		3.55
			IQR				IQR				IQR	
VAS pain (week)	*	5.85		2.5		5.76	±	1.51	*	6.45		2.6
			IQR								IQR	
**Cognitive status variables**												
Mini-mental	*	26		5	*	27		5	*	25.5		6
			IQR				IQR				IQR	
**Psychosocial-related variables**												
Beck depression inventory	*	12		7.5		10.33	±	5.99	*	12		8
			IQR								IQR	
VAMS anxiety		4.45	±	2.95		4.31	±	2.89		4.51	±	3.02
VAMS stress	*	3.35		5.07	*	3		5.9	*	3.35		4.8
			IQR				IQR				IQR	
VAMS depression	*	1.35		4.05	*	1.5		5.2	*	1.4		5.65
			IQR				IQR				IQR	
VAMS sleepiness	*	2.15		5.2	*	2.2		5.3	*	1.95		3.9
			IQR				IQR				IQR	
**Self-reported health-related quality of life**												
SF-12 physical component score		32.7	±	8.53		33.81	±	5.85	*	30.1		13.2
											IQR	
SF-12 mental component score		48.9	±	12.06		47.34	±	10.62		49.5	±	12.69
VAS global health assessment	*	4.25		3.5		4.3	±	2.12		4.69	±	2.64
			IQR									
**Disease-specific measures**												
Lequesne pain score		5.51	±	1.25		5.53	±	1.06		5.5	±	1.34
Lequesne max distance walked score	*	3		3.5		3.93	±	2.25	*	3		5
			IQR								IQR	
Lequesne ADLs score		4.61	±	1.45		4.4	±	1.62		4.69	±	1.39
Lequesne total score		13.8	±	3.92		13.86	±	3.71		13.7	±	4.06
WOMAC pain score		9.65	±	2.88		9.8	±	1.61		9.58	±	3.29
WOMAC rigidity score		3.41	±	1.98	*	4		2		3.38	±	2.08
							IQR					
WOMAC function score		35.1	±	11.88		32.4	±	9.6		36.3	±	12.65
WOMAC total score		48.2	±	14.91		45.66	±	10.74		49.3	±	16.35
**Performance-based physical function**												
One-leg stance test	*	4.4		8.76	*	5.03		5.81	*	2.67		7.22
			IQR				IQR				IQR	
Timed up and go test	*	15.5		9.13	*	14.96		9.02	*	16.3		12.62
			IQR				IQR				IQR	
**Quantitative sensory testing variables**												
Von Frey sensation—hand§	*	0.4	/	0.02–4	*	0.07			*	0.5	/	0.02–4
							IQR					
Von Frey pinprick—hand §	*	8	/	0.4–300	*	10			*	8	/	0.6–300
							IQR					
Von Frey pain threshold—hand §	*	300	/	6–300	*	300			*	300	/	6–300
							IQR					
Von Frey VAS pain—hand	*	1.33		3		1.34	±	1.86		1.71	±	1.85
			IQR									
Von Frey sensation—knee §	*	0.6	/	0.008–8	*	1			*	0.6	/	0.02–6
							IQR					
Von Frey pinprick—knee §	*	4	/	0.4–300	*	4			*	3	/	0.4–180
							IQR					
Von Frey pain threshold—knee§	*	26	/	6–300	*	15			*	26	/	2–300
							IQR					
Von-Frey VAS pain—knee		2.95	±	2.02		3.23	±	1.81		2.83	±	2.11
**Pain pressure threshold**												
Pain pressure threshold pre-CPM—hand	*	2.54		1.88		2.95	±	1.28	*	2.42		1.21
			IQR								IQR	
Pain pressure threshold pre-CPM—knee	*	1.77		1.38	*	1.5		1.76	*	1.65		1.08
			IQR				IQR				IQR	
Pain pressure threshold post-CPM—hand	*	2.41		1.58	*	2.36		1.63		2.55	±	1.13
			IQR				IQR					
Pain pressure threshold post-CPM—knee	*	2		1.46	*	2.03		1.69	*	2		1.29
			IQR				IQR				IQR	
**Conditioned pain modulation**												
Change in PPT after CPM—hand		3.56	±	27.68		−8.30	±	27.2		8.51	±	26.71
Percentage of change in pain after CPM—hand		1.44	±	22.22		1.2	±	19.05		1.53	±	23.66
Change in PPT after CPM—knee	*	12.3		28.55		23.57	±	24.9	*	14.2		25.84
			IQR								IQR	
Percentage of change in pain after CPM—knee	*	5		26.04		9	±	15.78	*	5.84		25.63
			IQR								IQR	

BPI, Brief Pain Inventory; *, non-normally distributed data (presented as the median and interquartile range (IQR); §, reported as the median, minimum, and maximum values; VAS, visual analog scale; now, at the moment of the assessment; week, average over the past week; VAMS, visual analog mood scale; ADLs, activity of daily living; hand and knee, test performed using the hand or the knee of the subject, respectively; CPM: conditioned pain modulation.

**Table 2 biomedicines-11-00004-t002:** Final predictive multivariate logistic model.

		95% CI			
Variable	OR	UB	LB	SE	*p*
VAMS sleepiness	38.41	1.5	989.16	63.67	0.028 *
Beck depression inventory	173.22	5.1	5887.15	311.62	0.004 **
SF-12 physical component score	59.07	2.5	1392.96	95.25	0.011 *
SF-12 mental component score	2585.38	18.2	367,878.60	6539.92	0.002 **
WOMAC total score	114.97	4.1	3258.17	196.17	0.005 **
Constant	1.18	0.00	8.76	5.71	0.001 **
Pseudo-R^2^	0.5611				
LR chi-squared (5)	34.67				0.000 **

Dependent variable: WOMAC pain subscale—success/non-success. N = 51; * *p* < 05; ** *p* < 01. OR, odds ratio; VAMS, visual analog mood scale for sleepiness.

**Table 3 biomedicines-11-00004-t003:** Accuracy for the five levels of the clinical prediction rule *.

Number of Predictor Variables Present	Sensitivity (95% CI)	Specificity (95% CI)	Positive Likelihood Ratio (95% CI)	Probability of Success, % (95% CI)	% Increase between Pre-Test and Post-Test Probability
5	0.80 (0.60–1.00)	0.94 (0.87–1.00)	14.40 (3.66–56.69)	85% (60–96%)	56%
4+	0.73 (0.51–0.96)	0.92 (0.83–1.00)	8.80 (2.86–27.12)	79% (54–92%)	50%
3+	0.53 (0.28–0.79)	0.94 (0.87–1.00)	9.60 (2.30–40.02)	80% (49–94%)	51%
2+	0.33 (0.09–0.57)	0.97 (0.92–1.00)	12.0 (1.53–94.23)	83% (39–98%)	54%

CI, confidence interval; +, or more. * The probability of success was calculated using the positive likelihood ratios and assumed a pre-test probability of 29%; total cases present, n = 51.

**Table 4 biomedicines-11-00004-t004:** The variables for the clinical prediction rule * and the number of subjects in each group at each level ^†^.

Number of Predictor Variables Present	Successful Outcome Group	Non-Successful Outcome Group
**5**	12	2
**4+**	11	3
**3+**	8	2
**2+**	5	1

*: VAMS sleepiness ≤2.9 (0–10), SF-12 physical component score ≤ 37.87, SF-12 mental component score ≤ 52.59, Beck depression inventory ≤10.5, and WOMAC total score ≤ 43.5. ^†^ n = 51. +, or more.

## Data Availability

The data that support the findings of this study are available from the corresponding author, Paulo E. P. Teixeira, upon reasonable request.

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
