# Peer review of "Development of a Clinical Prediction Rule for Treatment Success with Transcranial Direct Current Stimulation for Knee Osteoarthritis Pain: A Secondary Analysis of a Double-Blind Randomized Controlled Trial"

_biomedicines, 2022, doi:10.3390/biomedicines11010004_

Round 1
Reviewer 1 Report
The authors have tried to develop a clinical prediction rule to observe the analgesic effect of transcranial stimulation treatment on knee pain. They measured it using the WOMAC index and achieved a good success rate. The development of the clinical prediction rule has been carried out carefully and very successfully.
It is a study in which the authors should perform a series of minor revisions:
Abstract:
Please do not split the abstract into different points. The abstract should be a narrative of the objectives, material and methods, results, discussion, and conclusions. Eliminate the categorization of each of the sections.
Also, eliminate all acronyms included. Acronyms should be defined in the text itself and not in a section.
Introduction:
Please extrapolate KOA data to the world population, not just the U.S. population. Also, present the bibliography in Vancouver, please (mdpi uses its own Vancouver format).
Tables: please put everything in one size.
Author Response
Dear Reviewer,
Thank you for considering our manuscript for your assessment. We were encouraged by the comments made which recognized the strengths and impact of our work and we appreciated the constructive criticism, suggestions and clarifications asked.
Please find below a point-by-point response to your comments. We hope we have satisfied your requests and that you further consider our manuscript for publication. In view of your suggestions and requests, the manuscript was edited entirely, and the changes were highlighted. If there are any follow-up questions or clarifications, please don’t hesitate to reach out.
Abstract:
Please do not split the abstract into different points. The abstract should be a narrative of the objectives, material and methods, results, discussion, and conclusions. Eliminate the categorization of each of the sections.
Also, eliminate all acronyms included. Acronyms should be defined in the text itself and not in a section.
Response: Thank you for these observations and suggestions. We have edited the abstract accordingly.
Introduction:
Please extrapolate KOA data to the world population, not just the U.S. population. Also, present the bibliography in Vancouver, please (mdpi uses its own Vancouver format).
Response: Thank you for these comments. We have edited the first paragraph of the introduction to represent the incidence and prevalence of KOA worldwide and not only in the United States. We have also modified the references to the Vancouver format.
Tables: please put everything in one size.
Response: Thank you for the observation. We have edited all the table fonts to one size.
Reviewer 2 Report
Thanks for submitting the manuscript. The paper was well written. However, there are several issues need to be addressed by the authors:
1) In the introduction and methods sections, please provide more justifications for using CPR.
2) Did the authors obtained the longitudinal data? It will be more appealing to present the results with the baseline, immediately after the treatment, and follow up.
3) Please provide more justifications of the cutoff points, did the existing literature suggested any specific value(s) for each measures?
4) I have a bit concern about the sample size, there were only 51 recruited respondents, with only 15 successful cases. Please address this issue.
5) In view of the small sample size, the multivariate logistic model may have a potential danger of too many variables, but too few cases.
Author Response
Dear Reviewer,
Thank you for considering our manuscript for your assessment. We were encouraged by the comments made which recognized the strengths and impact of our work and we appreciated the constructive criticism, suggestions and clarifications asked.
Please find below a point-by-point response to your comments. We hope we have satisfied your requests and that you further consider our manuscript for publication. In view of your suggestions and requests, the manuscript was edited entirely, and the changes were highlighted. If there are any follow-up questions or clarifications, please don’t hesitate to reach out.
Reviewer comments:
1) In the introduction and methods sections, please provide more justifications for using CPR.
Response: Thank you for the comment. We have edited the 5th paragraph of the introduction to add more justification of why a CPR can be helpful for KOA.
2) Did the authors obtained the longitudinal data? It will be more appealing to present the results with the baseline, immediately after the treatment, and follow up.
Response: Thank you for this important comment. The objective of this study was to develop a CPR for the short-term, or immediately after, analgesic effect of tDCS. Therefore, for this particular study, we did not include the 2-month follow-up data. We have edited the last paragraph of the introduction, as well as the 2nd paragraph of the methods section to emphasized that the analysis was focused on the immediately after analgesic effect of tDCS. The follow-up data will be explored in a future submission.
3) Please provide more justifications of the cut-off points, did the existing literature suggested any specific value(s) for each measure?
Response: Thank you for this comment. The dichotomization of the independent variables is part of the methodology involved in the development of a CPR and is known to facilitate clinical understanding when interpreting the final prediction rule. During this process, we have considered the use of clinical cutoffs. However, as we explored several independent variables, it was difficult to find clear and well stablished clinical cutoffs for all the explored variables. In addition, according to our knowledge, most of previous research on development of CPRs has used the common ROC analysis statistical approach to define cutoffs for independent variables. Therefore, to standardize the procedure used for the calculation of the cutoffs, we have chosen to use the ROC curve analysis where we picked the best point (cutoff) that could better discriminate the outcome options (success or non-success). To clarify our methodology, we have edited the 2nd paragraph of the analysis section to clarify our choice on choosing the procedure for the dichotomization of the explored independent variables.
4) I have a bit concern about the sample size, there were only 51 recruited respondents, with only 15 successful cases. Please address this issue.
Response: Thank you for this comment. The current study is a secondary analysis of data from a published double-blind randomized controlled trial that evaluated whether tDCS reduces KOA pain in elderly individuals with a dysfunctional DPIS (NCT03117231). The parent trial included 104 subjects, however only 51 were randomized to the active tDCS groups. As our aim was to develop a CPR for the treatment of tDCS, we could only include these 51 patients who were randomized to the tDCS group. From those 51 patients, only 15 patients were classified on having had treatment success according to our definition (25% pain reduction from baseline on the WOMAC pain subscale). We understand that the use of a less strict criteria might would have captured more successful cases. However, we believe our criteria to determine treatment success was appropriate as it considered minimal detectable change, minimal clinically important difference, standard error of measurement, and minimum important change values previously reported for KOA. Therefore, the chosen criteria is likely to detect clinically meaningful improvements in pain which would strengthen the validity of the prediction rule. In addition, we believe that from the 51 patients, having approximately 30% classified as having treatment success was an adequate distribution that made the logistic model appropriate for the CPR development.
5) In view of the small sample size, the multivariate logistic model may have a potential danger of too many variables, but too few cases.
Response: Thank you for this comment. We do recognize this limitation in our limitations paragraph.
Reviewer 3 Report
I read this report with some interest. The authors reported the initial findings of the derivation phase of a clinical prediction rule for the treatment of knee OA pain with tDCS.
Specific comments:
1. As per the journal's instructions to authors, the abstract should be a total of about 200 words maximum, and it should follow the style of structured abstracts, but without headings.
2. Suggest to write 'knee OA' instead of using a nonstandard abbreviation like 'KOA'.
3. In terms of the population demographics, it would also be helpful to specify if the patients had single, bi- or tri-compartmental OA.
4. “… a total of seventeen variables were associated-12 with” – is there a typo here?
5. Why was the pretest probability of success significantly lower than that reported by Currier et al. (2007)? At least some explanation is necessary.
6. Please change “Womac TOTAL Score” to “WOMAC total score”.
7. Please change “Accuracy for the 5 levels” to “Accuracy for the five levels”.
8. Please change “Our finding goes with previous research” to “Our findings are similar to previous research”.
9. The relatively small sample size is a study limitation in itself. Overfitting is more likely with limited data points.
10. Please change “last session of the three weeks of treatment” to “last session after three weeks of treatment”.
11. Please provide a data availability statement.
Author Response
Dear Reviewer,
Thank you for considering our manuscript for your assessment. We were encouraged by the comments made which recognized the strengths and impact of our work and we appreciated the constructive criticism, suggestions and clarifications asked.
Please find below a point-by-point response to your comments. We hope we have satisfied your requests and that you further consider our manuscript for publication. In view of your suggestions and requests, the manuscript was edited entirely, and the changes were highlighted. If there are any follow-up questions or clarifications, please don’t hesitate to reach out.
Reviewer comments:
- As per the journal's instructions to authors, the abstract should be a total of about 200 words maximum, and it should follow the style of structured abstracts, but without headings.
Response: Thank you for this comment. We have edited the abstract accordingly.
- Suggest to write 'knee OA' instead of using a nonstandard abbreviation like 'KOA'.
Response: Thank you for this comment. We have edited the manuscript accordingly to use the standard abbreviation of knee OA.
- In terms of the population demographics, it would also be helpful to specify if the patients had single, bi- or tri-compartmental OA.
Response: Thank you for this comment. Indeed, to know how and where the knee joint is compromised could add valuable information. However, this data was not collected in the main parent trial and therefore we do not have it. We have edited our limitation paragraph (last paragraph of the discussion) to acknowledge this limitation.
- “… a total of seventeen variables were associated-12 with” – is there a typo here?
Response: Thank you for this observation. Yes, indeed this was a typo that was corrected.
- Why was the pretest probability of success significantly lower than that reported by Currier et al. (2007)? At least some explanation is necessary.
Response: Thank you for this observation. Differences in the prevalence of the rule, or in the pretest probability, is likely due to the differences in how treatment success was defined between the two studies (ours and Currier’s), and the unique characteristic of our sample of having an impaired descending pain inhibitory system. We have edited the 5th paragraph of the discussion to provide some explanation, as suggested.
- Please change “Womac TOTAL Score” to “WOMAC total score”.
Response: Thank you for this observation. We have reviewed the manuscript and edited this typo when appropriate.
- Please change “Accuracy for the 5 levels” to “Accuracy for the five levels”.
Response: Thank you for the observation. We have edited the noted observation accordingly.
- Please change “Our finding goes with previous research” to “Our findings are similar to previous research”.
Response: Thank you for the observation. We have edited the noted observation accordingly.
- The relatively small sample size is a study limitation in itself. Overfitting is more likely with limited data points.
Response: Thank you for the comment. We agree with the noted and we have edited our limitation paragraph to mention the relatively small sample size as a limitation.
- Please change “last session of the three weeks of treatment” to “last session after three weeks of treatment”.
Response: Thank you for the observation. We have edited the noted observation accordingly.
- Please provide a data availability statement.
Response: Thank you for the suggestion. We have added the data availability statement.
Reviewer 4 Report
The authors have developed an interesting secondary analysis of a double-blind randomized controlled trial developing a clinical prediction rule predicting a clinically significant analgesic effect on chronic knee osteoarthritis pain following transcranial direct current stimulation treatment.
However, I suggest several clarifications and modifications that will in my opinion improve the quality of their manuscript:
1. I suggest that authors include the research model in the title of their work.
2. In the introduction, I recommend that the authors to comment on the conservative treatment that can be very useful in mild to moderate OA. I recommend using and mentioning the following quality papers of studies investigating the use of Dry Needling and a 3-month program of Therapeutic Exercise, and Pain Education: doi:10.1093/pm/pnz036 , DOI: 10.3390/ijerph19106194
3. Please, could the authors square the table 1? In addition, the titles are missing in the heading
4. Please, could the authors square the table 4?
5. Given the nature of their work and their findings on signs of altered central nervous system processing and impaired pain modulation, I recommend that the authors include the following work for brief comment in the Discussion section: DOI: 10.3390/app11041895
6. Could the authors really summarize their conclusions? It feels like reading part of the introduction, as well as the limitations of the study.
Author Response
Dear Reviewer,
Thank you for considering our manuscript for your assessment. We were encouraged by the comments made which recognized the strengths and impact of our work and we appreciated the constructive criticism, suggestions and clarifications asked.
Please find below a point-by-point response to your comments. We hope we have satisfied your requests and that you further consider our manuscript for publication. In view of your suggestions and requests, the manuscript was edited entirely, and the changes were highlighted. If there are any follow-up questions or clarifications, please don’t hesitate to reach out.
Reviewer comments:
- I suggest that authors include the research model in the title of their work.
Response: Thank you for the comment and suggestion. We have edited the title of manuscript to clarify the that our research design was a secondary analysis of a double-blind randomized controlled trial.
- In the introduction, I recommend that the authors to comment on the conservative treatment that can be very useful in mild to moderate OA. I recommend using and mentioning the following quality papers of studies investigating the use of Dry Needling and a 3-month program of Therapeutic Exercise, and Pain Education: doi:10.1093/pm/pnz036 , DOI: 10.3390/ijerph19106194
Response: Thank you for this observation and suggestion. Indeed, we are aware of the benefits of the conservative treatment approach such as patient education in mild to moderate knee OA, and the suggested papers are excellent examples to highlight this. We have edited the introduction to cite the suggested papers, in order to exemplify other non-pharmacological treatment approaches with effect on pain perception.
- Please, could the authors square the table 1? In addition, the titles are missing in the heading
Response: Thank you for the observation and suggestion. We have edited table 1 to include the proper heading and the justified (square) formatting.
- Please, could the authors square the table 4?
Response: Thank you for the observation and suggestion. We have edited table 4 to the justified (square) formatting.
- Given the nature of their work and their findings on signs of altered central nervous system processing and impaired pain modulation, I recommend that the authors include the following work for brief comment in the Discussion section: DOI: 10.3390/app11041895.
Response: Thank you for the comment and suggestion. We have edited the 4th paragraph of the discussion to briefly comment on the published research that shows that motor-based interventions, such as mobilization with movement techniques, or manual therapy, can affect pain threshold and change pain mechanistic measures, and we have cited the suggested paper, which is a proper example of this evidence.
- Could the authors really summarize their conclusions? It feels like reading part of the introduction, as well as the limitations of the study.
Response: Thank you for the comment. We have edited the conclusion paragraph to summarize even more our findings and conclusions, following the commented. We believe it that now the conclusion is more direct and summarized. Again, thank you for the suggestion.
Round 2
Reviewer 2 Report
Thanks for submitting the revised manuscript to address my concerns. I am fully satisfied with the responses and changes made by the authors. Well done!
Reviewer 3 Report
Thank you for the revisions.
Reviewer 4 Report
The authors have improved the previous version of their work, and I consider recommend the current version for publication.